DATA RELEASE

# Online catalogue of the Coleção de Flebotomíneos (FIOCRUZ/COLFLEB), a biological collection of American sand flies (Diptera: Psychodidae, Phlebotominae) held at Fiocruz Minas, Brazil

José Dilermando Andrade-Filho[1,2,*], Alanna Silva Reis[1,2], Carolina Cunha Monteiro[1,2] and Paloma Helena Fernandes Shimabukuro[1,2]

1 Coleção de Flebotomíneos (FIOCRUZ/COLFLEB), Instituto René Rachou, Fiocruz Minas Avenida Augusto de Lima, 1715 - Barro Preto, 30190009, Belo Horizonte, Brazil
2 Grupo de Estudos em Leishmanioses, Instituto René Rachou, Fiocruz Minas Avenida Augusto de Lima, 1715 - Barro Preto, 30190009, Belo Horizonte, Brazil

## ABSTRACT

The Coleção de Flebotomíneos ("Phlebotomine Collection"; FIOCRUZ/COLFLEB) held at Fiocruz Minas is a curated biological collection comprising approximately 80 000 individual specimens of 370 species of sand flies (Diptera: Psychodidae, Phlebotominae). These were mostly collected from the Americas in the last 80 years by entomologists interested in understanding and controlling the vector-borne disease leishmaniases. Since 2010, the metadata of each biological specimen held in FIOCRUZ/COLFLEB, including the back catalogue of those deposited in previous decades, has been digitized. Here, we present the resulting electronic catalog containing records for 72,624 of the specimens, including all of the available provenance information associated with each specimen. The catalog is published online through the speciesLink network and the Sistema de Informação sobre a Biodiversidade Brasileira (SiBBr).

**Subjects** Ecology, Biodiversity, Taxonomy

**Submitted:** 28 February 2022

\* Corresponding author. E-mail: colfleb@fiocruz.br

Preprint submitted at https://doi.org/10.1590/SciELOPreprints.3782

Included in the series: *Vectors of human disease series* (https://doi.org/10.46471/GIGABYTE_SERIES_0002)

## DATA DESCRIPTION

### Background

Biological collections are repositories of biodiversity housing specimens and their related information, which can be used in various areas of scientific research [1]. These collections are especially important for insects of medical and veterinary importance, where accurate taxonomic identification is necessary for understanding and intervening in the epidemiology of vector-borne diseases [2]. The "Coleção de Flebotomíneos" ("Phlebotomine Collection") is held at the Instituto René Rachou, Fiocruz Minas (FIOCRUZ/COLFLEB), a federal public health research institution in Belo Horizonte, Minas Gerais, Brazil. FIOCRUZ/COLFLEB is a biological collection of preserved sand flies (Diptera: Psychodidae, Phlebotominae). These small insects are of considerable medical and public health importance because their blood-feeding adult females transmit protozoan parasites of the genus *Leishmania*, the etiological agent of leishmaniases, in addition to other bacterial and

**Figure 1.** Interactive map of the georeferenced occurrences hosted by GBIF [12]. https://www.gbif.org/dataset/2a629a9a-38d1-496b-afbf-b4ff3b8fae60

viral pathogens, which infect both human and non-human vertebrates [3]. The insect specimens deposited in FIOCRUZ/COLFLEB come from many different research projects carried out over the last 80 years. These projects were undertaken in many different areas of leishmaniases transmission, as well as from wild environments where leishmaniasis has never been recorded. The deposited sand flies were usually collected using light traps and have been identified by highly trained specialists using available taxonomic keys [2, 4]. The specimens can support research in the areas of taxonomy and systematics [5–7], and their associated data can be used in ecological niche/species distribution modelling [8], among other applications [9].

The dataset reported here is the metadata for each individual sand fly specimen deposited in FIOCRUZ/COLFLEB since 1953. Our dataset has 57 fields, which for each individual sand fly specimen describes its (i) taxonomy (kingdom, phylum, class, order, family, genus, specificEpithet, infraspecificEpithet, scientificName, scientificNameAuthorship, taxonRank, vernacularName, typestatus); (ii) collection details, including the collectors (recordedBy), collection date, trapping method, trap identification number, collection site description (occurrenceRemarks, eventDate, eventTime, habitat, samplingProtocol, samplingEffort, eventRemarks); (iii) geolocation data (country, countryCode, stateProvince, county, Island, waterbody, locality, locationRemarks, decimalLongitude, decimalLatitude, georeferenceRemarks); and (iv) catalogue reference data (occurrenceID, catalogNumber, OtherCatalogNumbers). The associated data for each physical object in our collection has a paper card with many fields referring to specimen provider, location, quantity of specimens, and so on. Data from these cards have been digitized by a dedicated member of the collection staff since 2010. The data are provided in the Darwin Core format [10]. Our data are available in the Sistema de Informação sobre a Biodiversidade Brasileira (SiBBr), an online platform that integrates data and information about biodiversity and ecosystems. SiBBr is the Brazilian node of the Global Biodiversity Information Facility (GBIF) [11], an internationally recognized resource for collating biological occurrence data. Our dataset has been submitted to GBIF, and is publicly available for use by others there (see Figure 1) [12].



## Context

Phlebotomine sand flies are insects of medical importance because they are involved in the transmission of pathogens between human and non-human animals. Approximately 1,000 sand flies species have been described, of which 530 are known to occur in the neotropical and nearctic regions [13].

FIOCRUZ/COLFLEB was officially started in 1953 as part of the work of Professor Amilcar Vianna Martins and Alda Lima Falcão, in collaboration with the technician João Evangelista da Silva [14]. The number of specimens in the collection increased considerably during the 1960s and 1970s. In the last 20 years, fossil specimens, as well as voucher specimens from DNA-based barcoding studies and other epidemiological studies involving fieldwork, have continued to be deposited in the collection [7, 15].

FIOCRUZ/COLFLEB contains 922 type specimens belonging to 151 species, including holotypes, allotypes, paratypes, plesiotypes, cotypes, topotypes, homeotypes, syntypes and neotypes. FIOCRUZ/COLFLEB also has a diverse collection of fossil species of neotropical sand flies, currently comprising 47 ambers from the Dominican Republic, within which 162 sand flies of nine species are preserved. Additionally, FIOCRUZ/COLFLEB has over 700 voucher specimens deposited from continuing DNA barcoding studies.

Efforts to digitize the metadata of biological specimens held in FIOCRUZ/COFLEB have been ongoing since 2010. Among the biological collections of the various regional institutions that comprise Fiocruz [16], FIOCRUZ/COLFLEB has the most published online data. Our online catalog is also integrated into the speciesLink network [17] and the Sistema de Informação sobre a Biodiversidade Brasileira (SiBBr) [18].

## METHODS

The sand flies held in FIOCRUZ/COLFLEB are adults and were collected using various methods, including resting collection from artificial and natural surfaces using either mechanical or manual aspirators ("pooters"); human-landing capture unbaited and $CO_2$-baited CDC-like light traps; and Shannon traps. Collections were made in different environments, including domestic and peridomestic (i.e., houses and their surroundings in urban, suburban and rural areas) and relatively undisturbed sylvatic and wild area (e.g., remote forest areas). Our specimens come from all 27 states of Brazil, and 19 other different countries of the Americas.

The live-caught insects are permanently mounted on glass microscope slides and preserved in either Berlese medium or Canada Balsam, while the fossil specimens are kept in plastic vials. All types are labelled with color-coded labels.

## DATA VALIDATION AND QUALITY CONTROL

Insects were identified by experienced taxonomists using keys available in the literature: those of Galati [2] and Young and Duncan [4].

The dataset is in Darwin Core format [10], with 57 possible terms available. All mandatory fields are present and have undergone screening in the FIOCRUZ IPT (Integrated Publishing Toolkit) [18], which is the GBIF software to provide data through their network. Metadata fields are also available on the online pages.

## RE-USE POTENTIAL

The data associated with the biological specimens deposited in FIOCRUZ/COLFLEB are of importance because they (i) describe the distribution of sand flies in different parts of

Brazil, and (ii) have a good temporal coverage – some specimens date back to 1939, while the most recent deposit was recorded in 2021. These data can be used for many different applications, in different research areas, including taxonomy and systematics [5–7], and ecological niche modelling [8], among others [9], to vector control activities [19, 20].

## DATA AVAILABILITY

The data supporting this article are published through the FIOCRUZ – Oswaldo Cruz Foundation IPT [18] and are available under a CC0 waiver from GBIF [12].

## EDITOR'S NOTE

This paper is part of a series of Data Release articles working with GBIF and supported by TDR, the Special Programme for Research and Training in Tropical Diseases, hosted at the World Health Organization [21].

## DECLARATIONS
## LIST OF ABBREVIATIONS

FIOCRUZ/COLFLEB: Coleção de Flebotomíneos, Fiocruz Minas ("Phlebotomine Collection of Fiocriz Minas"); GBIF: Global Biodiversity Information Facility; IPT: Information Publishing Tool; SiBBr: Sistema de Informação sobre a Biodiversidade Brasileira ("Information System for Brazilian Biodiversity"); TDR: the Special Programme for Research and Training in Tropical Diseases.

## ETHICAL APPROVAL

Not applicable.

## CONSENT FOR PUBLICATION

Not applicable.

## COMPETING INTERESTS

The author(s) declare that they have no competing interests.

## FUNDING

FIOCRUZ/COLFLEB is maintained and funded by the Fundação Oswaldo Cruz. JDAF received research fellowships from CNPq (302701/2016-8) and funding from the Fundação de Amparo à Pesquisa do Estado de Minas Gerais (FAPEMIG; PPM-00792-18). PHFS received a research grant from FAPEMIG (PPM-00676-18).

## AUTHORS' CONTRIBUTIONS

JDAF: collection curator, provision of resources, revision of the manuscript; ASR: data curation; CCM: data curation; PHFS: collection curator, preparation of the manuscript.

## ACKNOWLEDGEMENTS

We would like to thank the Vice-Presidência de Pesquisa e Coleções Biológicas/Fundação Oswaldo Cruz (VPPCB) for technical and financial support; Dr. Manuela da Silva and Dr. Aline da Silva Soares Souto (VPPCB). Juliana Xavier Faustino for technical support between 2013 to 2018; all the past and present students and researchers, who contributed in various ways to the formation, expansion and maintenance of the collection since its

inception in the 1960s; and Luke Baton for his comments on draft versions of this manuscript. We also thank Clara Baringo Fonseca (RNP – Rede Nacional de Ensino e Pesquisa) for technical support with the IPT.

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
