## [Reviewer Report]

Reviewer name and names of any other individual's who aided in reviewer Leonard E. MunstermannDo you understand and agree to our policy of having open and named reviews, and having your review included with the published papers. (If no, please inform the editor that you cannot review this manuscript.)YesIs the language of sufficient quality?NoPlease add additional comments on language quality to clarify if needed
Whereas the information content is clear, it is often presented as lengthy phrases joined by conjunctions. In Engllish, These clauses or phrases will be sepaarated by a period (full stop). Occasional unnecessary wordiness was also encountered.Are all data available and do they match the descriptions in the paper? YesAdditional CommentsThis is a very important database for research in phlebotomine taxonomy, as well as for understanding the epidemiology of leishmaniasis in the Americas. The Fiocruz Minas laboratories are renown for its current and historical collections, its highly skilled phlebotomine taxonomists, and its welcoming atmosphere. Are the data and metadata consistent with relevant minimum information or reporting standards? See GigaDB checklists for examples <a href="http://gigadb.org/site/guide" target="_blank">http://gigadb.org/site/guide</a>YesAdditional CommentsIs the data acquisition clear, complete and methodologically sound?YesAdditional CommentsIs there sufficient detail in the methods and data-processing steps to allow reproduction?YesAdditional CommentsIs there sufficient data validation and statistical analyses of data quality? Not my area of expertiseAdditional CommentsIs the validation suitable for this type of data?YesAdditional CommentsAgain, I am unsure about "validation". Certainly the species indentifications (the most important data) are of the highest caliber.Is there sufficient information for others to reuse this dataset or integrate it with other data?YesAdditional CommentsAny Additional Overall Comments to the AuthorPlease break up your lengthy sentences with several clauses (including what are termed 'dangling modifiers' into shorter sentences that contain only closely related information. An example is given below:
ORIGINAL SENTENCE: [The “Coleção de Flebotomíneos” ( the “Collection of Phlebotomines” as it is called in Portuguese) –held at the Instituto René Rachou, Fiocruz Minas (FIOCRUZ/COLFLEB), a federal public health research institution in the city of Belo Horizonte, in the state of Minas Gerais in Brazil is a biological collection of sand flies (Diptera: Psychodidae, Phlebotominae) small insects of considerable medical and public health importance because their blood-feeding adult females transmit protozoan parasites of the genus Leishmania, the etiological agent of leishmaniases, in addition to other bacterial and viral pathogens, which infect both human and non-human vertebrates [3] ]
.A SUGGESTED REWRITE: The Collection of Phlebotomines (or “Coleção de Flebotomíneos” in Portuguese) is held at a federal public health institution—the Instituto René Rachou, Fiocruz Minas (FIOCRUZ/COLFLEB) located in the city of Belo Horizonte, Minas Gerais, Brazil. It consists of a preserved collection of small flies, the phlebotomine sand flies (Diptera, Psychodidae, Phlebotominae). They are of considerable medical and public health importance, because by means of the blood feeding habit of the adult females, these flies can transmit a variety of protozoan, bacterial and viral diseases. The most important of these Is human leishmaniasis, a protozoan of the genus Leishmania [3]. RecommendationMinor Revision

---

## [Reviewer Report]

Reviewer name and names of any other individual's who aided in reviewer no bodyDo you understand and agree to our policy of having open and named reviews, and having your review included with the published papers. (If no, please inform the editor that you cannot review this manuscript.)YesIs the language of sufficient quality?NoPlease add additional comments on language quality to clarify if needed
Auhors should improve itAre all data available and do they match the descriptions in the paper? YesAdditional CommentsAre the data and metadata consistent with relevant minimum information or reporting standards? See GigaDB checklists for examples <a href="http://gigadb.org/site/guide" target="_blank">http://gigadb.org/site/guide</a>YesAdditional CommentsIs the data acquisition clear, complete and methodologically sound?YesAdditional CommentsIs there sufficient detail in the methods and data-processing steps to allow reproduction?YesAdditional CommentsIs there sufficient data validation and statistical analyses of data quality? Not my area of expertiseAdditional CommentsIs the validation suitable for this type of data?YesAdditional CommentsIs there sufficient information for others to reuse this dataset or integrate it with other data?YesAdditional CommentsAny Additional Overall Comments to the AuthorRecommendationAccept

---

## [Reviewer Report]

Reviewer name and names of any other individual's who aided in reviewer EDUARDO A. REBOLLAR-TELLEZDo you understand and agree to our policy of having open and named reviews, and having your review included with the published papers. (If no, please inform the editor that you cannot review this manuscript.)YesIs the language of sufficient quality?YesPlease add additional comments on language quality to clarify if needed
The manuscript is short, but well-written.Are all data available and do they match the descriptions in the paper? YesAdditional CommentsThe manuscript and the database if published represent an invaluable referencie source to all those researchers working on leishmaniasis vectors.Are the data and metadata consistent with relevant minimum information or reporting standards? See GigaDB checklists for examples <a href="http://gigadb.org/site/guide" target="_blank">http://gigadb.org/site/guide</a>YesAdditional CommentsAccess to Fiocruz/COLFLEB - Coleção de Flebotomíneos had no problem when using the provided link.Is the data acquisition clear, complete and methodologically sound?YesAdditional CommentsYes, the manuscript explains sufficiently well how database was originated.Is there sufficient detail in the methods and data-processing steps to allow reproduction?YesAdditional CommentsTha manuscript is not an experimental design, so perhaps the comments do not apply to the above-question.Is there sufficient data validation and statistical analyses of data quality? YesAdditional CommentsNo additional commentsIs the validation suitable for this type of data?YesAdditional CommentsNo additional commentsIs there sufficient information for others to reuse this dataset or integrate it with other data?YesAdditional CommentsYes, the manuscript is clear on this issue of how the dataset was elaborated and its potential use in different research fields. Any Additional Overall Comments to the AuthorPublication of this manuscript will certainly be a valuable resource to other researchers and institutions and also, will allow the recognition of the importance of preserving insect collection and its associated datasets.RecommendationAccept

---

## [Reviewer Report]

Upload additional filesDRR-202202-06/form/DRR-202202-06_Data-Review-MAT.pdfReviewer name and names of any other individual's who aided in reviewer Mary Ann TuliDo you understand and agree to our policy of having open and named reviews, and having your review included with the published papers. (If no, please inform the editor that you cannot review this manuscript.)YesIs the language of sufficient quality?YesPlease add additional comments on language quality to clarify if needed
Are all data available and do they match the descriptions in the paper? NoAdditional CommentsNot all specimens are classified to species level, but the paper does not state they are so I think this is OK. They are all sand flies. 
Many of the 'locality' values in the GBIF download contain non-ascii characters which makes the value unreadable. 
Many of the specimens do not include GPS values. I initially thought that it was just the older records (<1980s) but this does not seem to be the case, thus probably reflects how data for those records were collected.Are the data and metadata consistent with relevant minimum information or reporting standards? See GigaDB checklists for examples <a href="http://gigadb.org/site/guide" target="_blank">http://gigadb.org/site/guide</a>YesAdditional CommentsIs the data acquisition clear, complete and methodologically sound?YesAdditional CommentsIs there sufficient detail in the methods and data-processing steps to allow reproduction?YesAdditional CommentsIs there sufficient data validation and statistical analyses of data quality? YesAdditional CommentsIs the validation suitable for this type of data?YesAdditional CommentsIs there sufficient information for others to reuse this dataset or integrate it with other data?YesAdditional CommentsAny Additional Overall Comments to the AuthorRecommendationAccept